

# The influence of parenting styles and coping strategies on anxiety symptoms in adolescents: a comparative study of groups with and without non-suicidal self-injury behavior

Lingjiang Liu[1,2], Xinhui Hu[1,2], Huabing Xie[3], Changzhou Hu[1,2], Dongsheng Zhou[1,2], Jie Zhang[1,2], Yangjian Kong[4] and Fang Cheng[1,2]

[1] Department of Psychiatry, Affiliated Kangning Hospital of Ningbo University, NingBo, China
[2] Department of Psychiatry, Ningbo Kangning Hospital, NingBo, China
[3] Department of General Medicine, Renmin Hospital of Wuhan University, Wuhan, China
[4] The Second People's Hospital of Yuhuan, Yuhuan, China

Corresponding authors
Yangjian Kong,
addd198304@sina.com
Fang Cheng, cheng-
fang198220@sina.com

## ABSTRACT

**Background**. During the COVID-19 pandemic, the global incidence of non-suicidal self-injury (NSSI) has been increasing year by year, especially among adolescents, and it is highly correlated with the level of anxiety among teenagers, particularly within Chinese cultural contexts where parenting styles significantly influence adolescent behavior.

**Objective**. This study examines the role of culturally-influenced parenting styles and adolescent coping strategies in relation to anxiety and NSSI behavior, aiming to clarify these multivariate interactions for better understanding and intervention.

**Methods**. A cross-sectional survey was conducted with NSSI-diagnosed adolescents from Affiliated Kangning Hospital, Ningbo, China, and healthy adolescents from urban schools. Data on background, parenting styles, coping strategies, and anxiety levels were analyzed using chi-square tests, independent sample $t$-tests, MANOVA, and regression analysis.

**Results**. Adolescents with NSSI reported higher anxiety levels, more punishment and interference, and less emotional warmth from their fathers compared to the non-NSSI group ($p < 0.001$). The NSSI group also relied more on emotion-oriented coping strategies, while the non-NSSI group favored problem-oriented approaches ($p < 0.001$). Multivariate analyses confirmed significant effects of NSSI behavior, parenting styles, and coping strategies on anxiety levels.

**Conclusion**. This study found that the anxiety levels of NSSI adolescents were significantly higher than those of non-NSSI adolescents in China, which was closely related to less emotional warmth from parents, more punishment and interference, and emotion-focused coping strategies. The results demonstrate that family environment and coping strategies play a critical role in NSSI behavior and anxiety levels, highlighting the importance of emphasizing emotional support and positive coping strategies in the prevention and intervention of NSSI behavior.

# INTRODUCTION

As an increasing number of studies demonstrate, a large proportion of adolescents are involved in non-suicidal self-injury (NSSI) to some extent at some point in their lives, leading to growing concerns about NSSI among this population (*Baetens et al., 2014*; *Castro & Kirchner, 2018*). In particular, the fifth edition of the Diagnostic and Statistical Manual of Mental Disorders (DSM-5) (*Baetens et al., 2014*) has redefined NSSI as a new diagnostic entity: repetitive (occurring on more than 5 days within a year) acts of directly altering one's bodily tissues in a manner not socially sanctioned, and without suicidal intent during engagement (*American Psychiatric Association, 2013*). Under the interference of various factors such as social and family, the age range of NSSI onset shows an expanding trend (*Plener et al., 2015*; *Gandhi et al., 2018*; *Whitlock et al., 2011*). Despite some scholars' assertions that there is currently no evidence to suggest that non-suicidal self-injury without suicidal intent can lead to suicide (*Kapur et al., 2013*; *Zetterqvist, 2015*), many researchers have attempted to explore the link between NSSI and suicidal behavior (*Andover et al., 2012*) and have extensively documented NSSI as a risk factor for suicidal attempts (*Wilkinson et al., 2011*). Therefore, further research on adolescent NSSI is crucially important, particularly in the context of multi-factor interactive interventions.

As a transitional period from childhood to adulthood, adolescents often experience significant changes in social roles and living environments, which means they are more likely to experience negative emotions such as anxiety, especially during the COVID-19 pandemic and post-pandemic era (*World Health Organization, 2020*). The clinical occurrence rate of anxiety symptoms among adolescents worldwide is about 20.5%. In addition, a survey of high school students in 21 provinces and autonomous regions in China showed that the prevalence rate of anxiety symptoms among students was 37.4% (*Zhou et al., 2020*). However, many etiological models suggest that parenting behaviors play a key role in the development of anxiety disorders in children and adolescents (*Hudson & Rapee, 2004*), and these models are supported by considerable empirical research (*Waite & Creswell, 2015*; *McLeod, Wood & Weisz, 2007*). Parenting style refers to the attitudes, goals, and emotional atmosphere that parents use to raise and educate their children, and remains relatively stable in different circumstances (*Darling & Steinberg, 1993*; *Gorostiaga et al., 2019*). Parenting styles are deeply influenced by cultural backgrounds, which play a significant role in adolescents' anxiety symptoms and NSSI behaviors. At the same time, the coping style of adolescents has become an increasingly important indicator of their psychological states, such as anxiety. Coping behaviors refer to conscious strategies that individuals use when facing specific stressful events (*JA, 1984*). Meanwhile, *Spirito & Williams (1988)* proposed three coping styles: active coping, avoidance coping, and passive coping. During the COVID-19 pandemic, parenting styles within the context of traditional Chinese culture had a significant impact on adolescents' anxiety coping strategies. Confucianism, which

emphasizes family harmony and parental authority, further intensified the psychological stress and anxiety experienced by adolescents during the pandemic (*Dvorsky, Breaux & Becker, 2020*). Therefore, since the outbreak of COVID-19, parenting styles and adolescents' coping strategies have shown varying degrees of influence on individuals' anxiety emotions under stress, especially within the context of traditional Chinese culture. Moreover, this influence has shown complex and multidimensional mechanisms in the context of NSSI.

Evidence suggests that parenting styles and coping behaviors have a significant impact on NSSI behaviors (*Cheng et al., 2022*). In fact, poor parenting styles are associated with increased risks of adolescent anxiety, hostility, and poor social adaptation (*Anosike, Isah & Igboeli, 2020*; *Martin et al., 2016*). Parenting styles may directly affect NSSI behaviors, but different parenting behaviors, such as overinvolvement and overreaction, may have different effects on NSSI behaviors. Currently, research has found that negative emotions such as anxiety are associated with NSSI behavior (*Selby et al., 2012*), and self-harm is often used to regulate these aversive emotions (*Turner, Chapman & Layden, 2012*). In terms of specific coping strategies, individuals involved in NSSI are significantly less likely to engage in problem-focused or emotion-focused coping, seek instrumental support, or engage in religious or spiritual forms of coping; instead, they are more likely to use substance abuse, behavioral disengagement, and self-blame as coping mechanisms (*Wester & Trepal, 2010*). In the above study, we found that parental parenting styles and adolescent coping behaviors both have varying degrees of impact on individuals' negative emotions such as anxiety. At the same time, NSSI is also related to parental parenting styles and adolescent coping styles, as well as psychological states such as anxiety. However, currently there are no scholars who have been able to deeply analyze the interactive impact mechanism of parental parenting styles and adolescent coping styles on individual anxiety in the diagnostic context of NSSI.

To address these gaps, this study aims to analyze the complex impact of parental parenting styles and adolescent coping styles on individual anxiety in both NSSI and non-NSSI groups. Specifically, we attempt to break the limitations of the single variable impact mechanism and propose a significance analysis method under the interference of multiple factors and interactions. Through this study, we hope to provide valuable references and guidance for adolescent mental health education and parenting strategies. This study focuses on how to develop effective prevention and intervention measures by understanding the impact of parental parenting styles and adolescent coping strategies on individual mental health. By studying the differences in impact between the NSSI and non-NSSI groups, we can develop targeted solutions to promote their overall psychological health.

## METHODS

### Study design

This study is a cross-sectional survey targeting adolescents and their parents from Ningbo, China. The parenting styles of the parents are deeply influenced by Confucian traditions, emphasizing parental authority and family harmony, which are considered key cultural background factors in the study. Given the increasing trend in non-suicidal self-injury

(NSSI) incidence rates, a questionnaire survey was conducted on NSSI patients admitted and diagnosed at the Kangning Hospital affiliated with Ningbo University in Zhejiang Province, China, between June 2020 and December 2021. Meanwhile, during this period, a data investigation and research was conducted by selecting three middle schools and three high schools in each of the four districts (Haishu, Jiangbei, High-tech Zone, and Yinzhou) in the urban areas of Ningbo, using frequency matching by gender and age in the case group. The survey employed a self-made general information questionnaire, a coping style scale for middle school students, a parenting style questionnaire, and an anxiety assessment scale. This study has been approved by the Ethics Committee of Kangning Hospital, Ningbo, Zhejiang Province, China (Ethics No. NBKNYY-2019-LC-10, 2019.7.25–2022.7.24). All participants were informed of the purpose and content of the survey and signed the digital informed consent form. All questionnaires were completed anonymously.

## Participants

The medical records group was selected from the Child and Adolescent Mental Health Department of Kangning Hospital, Ningbo, and enrolled patients who were hospitalized between June 2020 and December 2021, using a sequential inclusion method. The inclusion criteria were as follows: (1) meeting the diagnostic criteria for depression and non-suicidal self-injury (NSSI) in the fifth edition of the Diagnostic and Statistical Manual of Mental Disorders (DSM-5); (2) first-time visit and no previous treatment received; (3) aged between 11 and 18 years; (4) patients and their guardians provided informed consent for the study. Exclusion criteria included: (1) presence of severe physical or organic brain disorders; (2) absence of concurrent mental disorders other than anxiety and depression; (3) history of severe substance abuse. Ultimately, a total of 311 patients, including 31 males and 280 females, were enrolled.

The control group consisted of healthy adolescents from three middle schools and three high schools in the urban area of Ningbo city, selected by frequency matching of gender and age with the case group. According to the DSM-5 diagnostic criteria, the following are non-judgmental questions designed to define NSSI behavior: (1) intentional self-harm behavior that causes mild or moderate physical harm such as bleeding, bruising, or pain, occurring more than 5 times in the past year, and not to end life; (2) the behavior is aimed at relieving or solving difficult interpersonal communication or inducing positive emotional states from a state of tension or cognition. If both of the above questions are answered "yes", then it is determined as having NSSI behavior; otherwise, it is determined as not having NSSI behavior. In the end, a total of 311 adolescents without NSSI behavior were included, including 42 males and 269 females.

All enrolled study subjects were diagnosed by a chief psychiatrist, and were assessed and evaluated by trained attending psychiatrists with consistency after thorough explanation of the study objectives and significance. The test subjects independently completed the test by accurately filling out all items in the survey form. Trained personnel provided specific guidance for items that were difficult for the subjects to comprehend, but refrained from providing any leading prompts. All survey questionnaires were collected on-site.

## Survey content

Referring to the survey questionnaire in past studies (*Wan et al., 2015*), we developed a general information questionnaire consisting of three parts. The first part requests basic information, such as the child's gender and age.

The second part consists of the Parental Rearing Behavior Inventory (EMBU) and the Coping Style Questionnaire for High School Students. The questionnaire on parental rearing behaviors was developed by *Perris et al. (1980)*, introduced and revised by *Yue et al. (1993)* (*Wang, Wang & Ma, 1999*). It assesses parental attitudes and behaviors, consisting of 66 items, each rated on a 4-point scale: 1, never; 2, occasionally; 3, frequently; 4, always. The father's parental rearing style includes six factors: emotional warmth and understanding (F1), strict punishment (F2), refusal to deny (F3), preference for subjects (F4), excessive interference (F5), and overprotection (F6). The mother's parental rearing style includes five factors: emotional warmth and understanding (M1), strict punishment (M2), refusal to deny (M3), preference for subjects (M4), and excessive interference/overprotection (M5). The Coping Style Questionnaire for High School Students was developed by *Chen et al. (2000)* It includes 36 items, each rated on a 4-point scale: 1, not used; 2, used occasionally; 3, used sometimes; 4, used frequently. It includes two subscales: the problem-oriented coping subscale, consisting of three factors: problem solving (P1), seeking social support (P2), and positive reinterpretation (P3); and the emotion-oriented coping subscale, consisting of four factors: endurance (E1), avoidance (E2), venting (E3), and fantasy denial (E4).

The last section is the Screen for Child Anxiety Related Emotional Disorders (SCARED), which was developed by Birmaher. *Wang, Su & Zhu (2002)* validated the reliability and validity of this questionnaire in China and established a norm for urban Chinese children and adolescents aged 8-16 years to self-assess their own anxiety. The questionnaire comprises 41 items, each rated on a 3-point scale (0 indicating the absence of symptoms, 1 indicating the presence of some symptoms, and 2 indicating the frequent presence of symptoms). It includes five factors, namely generalized anxiety (A1), physical symptoms/panic (A2), separation anxiety (A3), social phobia (A4), and school phobia (A5). A total score of 23 or higher indicates the presence of anxiety. The authors have permission to use these instruments from the copyright holders.

To ensure the reliability and consistency of the questionnaire results, the psychiatrist explained the purpose of the survey and managed the administration of the questionnaire to the participants A pre-test was conducted before the formal investigation. After completing the questionnaire, the doctor in charge collected the questionnaires, entered and screened the two copies, and then checked them to ensure the quality of the answers.

## Data analysis methods

After logic checking and proofreading, we used R−4.2.1 (*R Core Team, 2022*) and RStudio for windows (an open-source IDE) to process and analyze the data. We used chi-square test and independent sample $t$-test to compare the intergroup differences of survey variables among different groups of adolescents respectively (Table 1). By comparing the gender and age of NSSI group and non-NSSI group adolescents, we can demonstrate the effectiveness of frequency matching sampling. Demonstrating the significant differences in perception

**Table 1  Single-factor analysis of variables.**

| Variables | Sub-variables | Non-NSSI Count/Mean (%/SD) | NSSI Count/Mean (%/SD) | Z-Value | p-Value |
|---|---|---|---|---|---|
| Gender | Male | 42 (13.5) | 31 (10.0) | 1.55 | 0.213 |
| | Female | 269 (86.5) | 280 (90.0) | | |
| Age | | 14.08 (1.31) | 14.49 (1.51) | 4.67 | 0.091 |
| SCARED | A1 | 9.38 (8.70) | 14.34 (6.71) | 11.9 | <.001*** |
| | A2 | 8.73 (5.34) | 12.81 (4.17) | 11.03 | <.001*** |
| | A3 | 6.92 (4.89) | 6.97 (3.55) | 12.17 | 0.881 |
| | A4 | 7.10 (4.26) | 8.76 (3.33) | 12.44 | <.001*** |
| | A5 | 2.87 (2.86) | 5.51 (2.16) | 6.33 | <.001*** |
| Anxiety Score | | 35.00 (24.14) | 48.40 (16.24) | 17.34 | <.001*** |
| Father's parenting style | F1 | 50.38 (13.29) | 37.81 (9.56) | 13.25 | <.001*** |
| | F2 | 19.68 (7.16) | 21.94 (9.19) | 8.47 | .001** |
| | F3 | 20.85 (5.40) | 11.32 (4.15) | 7.12 | <.001*** |
| | F4 | 10.40 (4.64) | 14.13 (6.53) | 5.1 | <.001*** |
| | F5 | 10.44 (3.72) | 21.97 (6.31) | 6.74 | <.001*** |
| | F6 | 12.74 (3.30) | 10.60 (3.23) | 7.8 | <.001*** |
| Mother's parenting style | M1 | 51.44 (13.09) | 42.07 (10.39) | 14.36 | <.001*** |
| | M2 | 36.99 (8.19) | 16.33 (6.08) | 8.17 | <.001*** |
| | M3 | 14.82 (5.30) | 16.58 (5.72) | 10.85 | <.001*** |
| | M4 | 14.36 (5.72) | 14.37 (6.34) | 4.64 | 0.979 |
| | M5 | 10.56 (4.65) | 39.74 (9.15) | 6.84 | <.001*** |
| Problem-solving–oriented coping | P1 | 20.57 (4.97) | 14.81 (4.54) | 9.99 | <.001*** |
| | P2 | 19.19 (4.56) | 14.95 (4.27) | 9.73 | <.001*** |
| | P3 | 14.03 (3.67) | 10.05 (3.32) | 8.53 | <.001*** |
| Emotional-oriented coping | E1 | 9.92 (2.46) | 11.05 (2.37) | 6.61 | <.001*** |
| | E2 | 8.03 (2.55) | 9.89 (2.67) | 7.4 | <.001*** |
| | E3 | 9.00 (2.79) | 9.81 (2.87) | 9.17 | <.001*** |
| | E4 | 10.77 (3.75) | 13.05 (3.80) | 12.57 | <.001*** |

**Notes.**

SCARED,  Screen for Child Anxiety Related Emotional Disorders.

***$p < .001$

**$p < .01$

Z-Value and p-Value represent the statistical significance of differences between groups.

and behavior between different adolescent groups, we compared parental rearing styles (F1–F6 for fathers, M1–M5 for mothers), coping behaviors (P1–P3 for problem-solving orientation, E1–E4 for emotion-oriented coping strategies), and anxiety scores.

Next, we used multivariate analysis of variance (MANOVA) to explore the overall impact of the independent variable on a set of dependent variables, which in this case refers to multiple measurement indicators related to adolescent anxiety. The use of MANOVA allows us to evaluate whether each independent variable has an overall significant effect on a set of related dependent variables, and this step is crucial for subsequent analysis. In this way, we can not only detect the impact of individual independent variables, but also discover the possible interaction effects among multiple independent variables, providing us with a global perspective to understand how independent variables interact with a set

**Table 2  Multivariate analysis of variance (MANOVA) results for the impact of various variables on anxiety measures by NSSI group status.**

| Variables | Pillai's Trace | *F*-Value | Numerator df | Denominator df | *p*-Value |
|---|---|---|---|---|---|
| NSSI Group Status | 0.424 | 88.04 | 5 | 598 | <.0001*** |
| F1 | 0.192 | 28.50 | 5 | 598 | <.0001*** |
| F2 | 0.053 | 6.72 | 5 | 598 | <.0001*** |
| F3 | 0.061 | 7.78 | 5 | 598 | <.0001*** |
| F4 | 0.075 | 9.75 | 5 | 598 | <.0001*** |
| F5 | 0.044 | 5.49 | 5 | 598 | <.0001*** |
| F6 | 0.130 | 17.89 | 5 | 598 | <.0001*** |
| M1 | 0.043 | 5.36 | 5 | 598 | <.0001*** |
| M2 | 0.041 | 5.13 | 5 | 598 | <.0001*** |
| M3 | 0.010 | 1.22 | 5 | 598 | .298 |
| M4 | 0.017 | 2.04 | 5 | 598 | .071 |
| M5 | 0.027 | 3.29 | 5 | 598 | .006** |
| E1 | 0.066 | 8.47 | 5 | 598 | <.0001*** |
| E2 | 0.118 | 16.01 | 5 | 598 | <.0001*** |
| E3 | 0.037 | 4.57 | 5 | 598 | <.0001*** |
| E4 | 0.046 | 5.81 | 5 | 598 | <.0001*** |
| P1 | 0.091 | 12.02 | 5 | 598 | <.0001*** |
| P2 | 0.023 | 2.80 | 5 | 598 | .016* |
| P3 | 0.009 | 1.10 | 5 | 598 | .357 |

**Notes.**
*$p < .05$
**$p < .01$
***$p < .001$

of dependent variables (see Table 2 for the multivariate analysis of variance (MANOVA) results, which show an overall view of the effects of various variables on adolescent anxiety measures and distinguish the NSSI group status).

After identifying the independent variables with significant effects through MANOVA, we used regression analysis of main effects and NSSI interaction terms to further explore the relationship between these variables and specific anxiety factors. These regression models considered the main effects between parental childrearing practices, adolescent coping strategies, anxiety levels, and the potential interaction effects between these variables and NSSI status. Through this method, we are able to specifically understand how parents' specific behaviors and adolescent coping mechanisms independently and interactively impact anxiety levels, while also revealing the differences in these relationships between the NSSI and non-NSSI groups (see Table 3 for a summary of the regression analysis results across the adolescent anxiety subscales A1-A5). The purpose of this analysis is to better understand and predict the factors influencing adolescent anxiety, and how they operate differently across different groups, in order to provide data support for future prevention and intervention strategies.

All results were considered statistically significant at $p < 0.05$. In cases of multiple comparisons, we applied Bonferroni correction to adjust the significance level and control for Type I error (*i.e.,* false positives). In conclusion, we used a combination of chi-square

**Table 3   Summary of regression analysis results across anxiety subscales A1–A5.**

| Predictor | A1 Estimate (p) | A2 Estimate (p) | A3 Estimate (p) | A4 Estimate (p) | A5 Estimate (p) |
|---|---|---|---|---|---|
| Intercept | 3.51 (0.317) | 0.7 (0.748) | 3.27 (0.111) | 3.83 (0.041$^*$) | 1.38 (0.239) |
| Group-NSSI | −6.6 (0.167) | −0.84 (0.779) | −4.08 (0.145) | 1.03 (0.686) | 2.56 (0.109) |
| F1 | −0.07 (0.228) | −0.08 (0.026$^*$) | −0.06 (0.072) | −0.08 (0.015$^*$) | −0.05 (0.019$^*$) |
| F2 | −0.1 (0.208) | −0.04 (0.379) | −0.09 (0.058) | −0.06 (0.184) | −0.01 (0.694) |
| F3 | 0.03 (0.768) | 0.03 (0.615) | −0.03 (0.584) | −0.05 (0.331) | 0 (0.907) |
| F4 | 0.3 (0.123) | 0.29 (0.017$^*$) | 0.17 (0.146) | 0.17 (0.096) | 0.06 (0.397) |
| F5 | −0.08 (0.631) | −0.09 (0.381) | −0.02 (0.872) | −0.05 (0.578) | −0.05 (0.312) |
| F6 | 0.25 (0.081) | 0.23 (0.01$^{**}$) | 0.25 (0.003$^{**}$) | 0.23 (0.003$^{**}$) | 0.1 (0.034$^*$) |
| M1 | −0.04 (0.504) | −0.01 (0.884) | 0 (0.917) | −0.01 (0.748) | 0 (0.928) |
| M2 | 0.21 (<0.001$^{***}$) | 0.15 (<0.001$^{***}$) | 0.1 (0.007$^{**}$) | 0.1 (0.003$^{**}$) | 0.05 (0.023$^*$) |
| M5 | 0.17 (0.387) | −0.09 (0.458) | 0.06 (0.575) | −0.03 (0.785) | 0.08 (0.245) |
| E1 | 0.2 (0.199) | 0.16 (0.105) | 0.13 (0.143) | 0.1 (0.21) | 0.02 (0.771) |
| E2 | 0.28 (0.104) | 0.09 (0.423) | 0.16 (0.118) | 0.03 (0.771) | 0.19 (<0.001$^{***}$) |
| E3 | −0.01 (0.919) | 0.14 (0.126) | −0.04 (0.655) | 0.07 (0.357) | 0.11 (0.025$^*$) |
| E4 | 0.44 (<0.001$^{***}$) | 0.3 (<0.001$^{***}$) | 0.26 (<0.001$^{***}$) | 0.26 (<0.001$^{***}$) | 0.1 (0.01$^{**}$) |
| P1 | −0.38 (<0.001$^{***}$) | −0.12 (0.078) | −0.26 (<0.001$^{***}$) | −0.18 (0.003$^{**}$) | −0.07 (0.049$^*$) |
| P2 | −0.17 (0.103) | −0.08 (0.251) | 0.02 (0.711) | 0.06 (0.286) | −0.11 (0.001$^{***}$) |
| Group-NSSI: F1 | 0.11 (0.149) | 0.12 (0.014$^*$) | 0.1 (0.03$^*$) | 0.1 (0.016$^*$) | 0.03 (0.308) |
| Group-NSSI: F2 | 0.12 (0.279) | 0.07 (0.333) | 0.19 (0.004$^{**}$) | 0.08 (0.198) | 0.01 (0.738) |
| Group-NSSI: F3 | −0.03 (0.859) | 0.05 (0.649) | −0.04 (0.713) | 0.2 (0.038$^*$) | −0.02 (0.693) |
| Group-NSSI: F4 | −0.3 (0.144) | −0.3 (0.019$^*$) | −0.21 (0.077) | −0.23 (0.034$^*$) | −0.08 (0.233) |
| Group-NSSI: F5 | 0.26 (0.163) | 0.07 (0.514) | −0.08 (0.439) | −0.04 (0.685) | 0.07 (0.265) |
| Group-NSSI: F6 | 0.01 (0.968) | 0.04 (0.786) | −0.02 (0.888) | −0.08 (0.46) | 0.01 (0.925) |
| Group-NSSI: M1 | 0.06 (0.417) | 0.04 (0.409) | 0.06 (0.159) | 0.03 (0.389) | 0 (0.881) |
| Group-NSSI: M2 | −0.02 (0.853) | −0.17 (0.016$^*$) | −0.13 (0.057) | −0.12 (0.053) | −0.06 (0.142) |
| Group-NSSI: M5 | −0.11 (0.574) | 0.15 (0.245) | 0.01 (0.934) | 0.07 (0.517) | −0.05 (0.478) |
| Group-NSSI: E1 | 0 (0.995) | 0.11 (0.457) | −0.16 (0.253) | 0 (0.97) | 0.05 (0.537) |
| Group-NSSI: E2 | −0.17 (0.48) | 0.06 (0.664) | −0.04 (0.777) | −0.09 (0.5) | −0.09 (0.251) |
| Group-NSSI: E3 | 0.14 (0.493) | 0.01 (0.942) | 0.09 (0.449) | −0.04 (0.687) | −0.04 (0.529) |
| Group-NSSI: E4 | −0.16 (0.305) | −0.16 (0.1) | −0.13 (0.176) | −0.15 (0.067) | −0.1 (0.051) |
| Group-NSSI: P1 | 0.08 (0.599) | −0.1 (0.283) | 0.14 (0.128) | −0.04 (0.626) | −0.03 (0.561) |
| Group-NSSI: P2 | 0.09 (0.581) | 0.06 (0.555) | −0.1 (0.278) | −0.02 (0.78) | 0.1 (0.063) |

**Notes.**
$^*p < .05$
$^{**}p < .01$
$^{***}p < .001$

tests, t-tests, MANOVA and regression analysis that considered main and interactive effects to comprehensively evaluate the relationships between adolescent non-suicidal self-injury (NSSI) behavior and parenting styles, adolescent coping strategies, and anxiety levels. Our analysis focused on exploring how different parenting styles and adolescent coping strategies affect adolescent anxiety symptoms, with special attention to examining the differences in these effects between NSSI and non-NSSI adolescent groups. The aim of this study is to identify key family and individual factors that may either decrease or increase

the risk of adolescent NSSI, thereby providing theoretical and practical guidance for the development of prevention and intervention strategies.

## RESULTS

### Single factor intergroup comparative analysis

Table 1 (univariate analysis) explored the differences between the non-NSSI group and the NSSI group in various variables. The results showed that females accounted for a large proportion in both groups in terms of gender distribution (86.5% in the non-NSSI group and 90% in the NSSI group). With regard to age, the NSSI group and non-NSSI group were aged $14.49 \pm 1.51$ and $14.08 \pm 1.31$, respectively. However, there was no statistically significant correlation between gender and age with NSSI behavior ($Z = 1.55$, $p = 0.213$; $Z = 4.67$, $p = 0.091$). This result demonstrates the effectiveness of frequency matching when selecting the control group, which effectively controlled for the influence of the two basic variables of age and gender on subsequent analyses.

In the detailed assessment of the SCARED anxiety scale, apart from separation anxiety (A3), adolescents with NSSI behavior had significantly higher scores than the non-NSSI group in all dimensions, including generalized anxiety (A1), somatic/panic (A2), social anxiety (A4), and school phobia (A5), and all these differences were highly statistically significant ($p < 0.001$). This result clearly indicates that adolescents in the NSSI group exhibited a significant increase in multiple anxiety areas, revealing that they face greater challenges in a broad range of anxiety issues. Meanwhile, the overall anxiety score of the NSSI group was significantly higher than that of the non-NSSI group ($p < 0.001$), which further confirms the close link between NSSI behavior and higher levels of anxiety emotions. These findings emphasize the importance of anxiety in the context of adolescent NSSI behavior, indicating that these psychopathological states are not only marked features of NSSI behavior but may also be crucial psychological driving factors contributing to the onset and maintenance of NSSI behavior.

Some significant differences were found between the NSSI and non-NSSI groups in terms of parenting styles on multiple dimensions. Notably, significant differences were observed between the NSSI and non-NSSI groups on subscales F1 to F6 related to paternal parenting style. Additionally, the non-NSSI group scored significantly higher than the NSSI group on F1 (Emotional Warmth and Understanding), F3 (Rejection and Denial), and F6 (Overprotection), indicating that adolescents in the non-NSSI group may have felt more love and protection from their fathers. In contrast, the NSSI group scored higher on F2 (Punishment and Severity), F4 (Preference for the Subject), and F5 (Excessive Interference) compared to the non-NSSI group. These results suggest that adolescents with NSSI may feel a lack of emotional warmth and understanding from their fathers, while also experiencing excessive punishment and interference. Similarly, analysis of maternal parenting styles revealed a similar pattern, with significant score differences between the NSSI and non-NSSI groups on variables M1 to M5, except for M4 (Preference for the Subject). The non-NSSI group scored higher on M1 (Emotional Warmth and Understanding) and M2 (Punishment and Severity), while the NSSI group showed significantly higher scores on M3

(Rejection and Denying) and M5 (Excessive Interference and Protection). This suggests that adolescents in the NSSI group may perceive more interference and protection from their mothers instead of emotional warmth and understanding.

In terms of coping styles in adolescents, problem-solving coping (P1, P2, P3) was significantly higher in the non-NSSI group than in the NSSI group (all $p < 0.001$), while emotion-focused coping (E1, E2, E3, E4) was significantly higher in the NSSI group than in the non-NSSI group (all $p < 0.001$). These results indicate that adolescents with NSSI behavior tend to adopt more emotion-focused coping strategies rather than problem-solving-focused strategies. These differences highlight the potential impact of adolescent coping strategy choices on their mental health and behavioral patterns, especially in dealing with stress and emotional challenges. Problem-solving coping strategies may help adolescents manage stress more effectively and reduce the risk of self-harm behavior, while emotion-focused coping strategies may be associated with higher levels of emotional distress and self-harm behavior risk.

The above results indicate that compared to non-NSSI individuals, NSSI individuals score lower in certain parenting styles and problem-solving coping strategies, but higher in emotion-focused coping strategies and anxiety measures, suggesting that the differences in anxiety levels between individuals with NSSI behavior and those without may be related to parenting styles and adolescent coping strategies.

## Multivariate analysis of variance (MANOVA) results

In the second part of this study, we used multivariate analysis of variance (MANOVA) to investigate the overall impact of NSSI group status and other variables (including parenting styles and adolescent coping strategies) on adolescent anxiety scale scores. The MANOVA results revealed that NSSI group status and most independent variables had a significant overall impact on anxiety measures, providing us with important insights into how different variables interact to affect adolescent anxiety.

Specifically, the Pillai's Trace value for NSSI Group Status was 0.424, with an F value of 88.04, indicating a significant difference ($p < .0001***$) in anxiety levels between individuals with and without NSSI behavior, highlighting the important impact of NSSI behavior on adolescent anxiety.

In terms of parenting styles, the analysis results for each sub-variable (F1 to F6 representing father's parenting styles, M1 to M5 representing mother's parenting styles) also showed significant statistical significance (all $p$-values were $<.0001***$), except for M3 ($p= .298$) and M4 ($p= .071$). This finding indicates that the majority of dimensions of parenting styles have an overall effect on adolescent anxiety levels, and this effect is statistically significant.

Furthermore, most of the variables in adolescent coping strategies (P1 to P3 representing problem-solving-oriented coping strategies, E1 to E4 representing emotion-oriented coping strategies) also showed significant overall effects (all $p$-values $<.0001***$), except for P2 ($p= .016*$) and P3 ($p= .357$). This further confirms that adolescent coping strategies play an important role in their levels of anxiety.

These MANOVA results highlight the overall impact of NSSI behavior, parenting styles, and adolescent coping strategies on adolescent anxiety levels, revealing the importance of considering these variables in NSSI research. These results not only highlight the joint impact of different factors on adolescent anxiety, but also provide a solid foundation for further analyzing the specific reasons for the difference in anxiety levels between NSSI and non-NSSI groups. Additionally, this has screened variables with significant impact for subsequent regression analyses of main effects and NSSI interaction terms, providing a foundation for exploring how particular variables affect adolescent anxiety levels and how these relationships vary across different groups.

## Regression analysis of main effects and NSSI interaction terms

In this section, we focus on the regression analysis results of main effects and NSSI interaction terms, which aim to explore in-depth how parenting styles, adolescent coping strategies, and NSSI behaviors affect adolescent anxiety levels, and examine the specific effects of interactive factors on different anxiety subscales (A1–A5).

The results of the regression analysis indicate that multiple dimensions of adolescent anxiety are significantly affected by various factors, including NSSI group status, parenting styles (F1–F6, M1–M5), and adolescent coping strategies (P1–P3, E1–E4). Specifically, for father's parenting style, the lower perceived warmth of F1 (emotional warmth, understanding) is associated with higher anxiety scores, particularly in A2 (somatic/panic), A4 (social phobia), and A5 (school phobia). The estimated value of F4 (favoritism of the subject) on A2 (somatic/panic) suggests that higher levels of indulgence favoritism are related to higher levels of anxiety. The significant positive effect of F6 (overprotection) on A2–A5 indicates that overprotection is correlated with high levels of anxiety. Regarding mother's parenting style: M2 (harsh punishment) significantly increases the likelihood of anxiety on all anxiety subscales, especially on A1 (generalized anxiety) and A2 (somatic/panic), indicating that mother's harsh parenting style may further increase adolescent's anxiety Regarding adolescent coping strategies (E1–E4, P1–P2): The significant positive estimated value of E4 (emotion-focused coping strategies) on all anxiety subscales reveals that the more adolescents rely on emotion-focused coping strategies, the higher their anxiety levels may be. On the contrary, the significant negative estimated value of P1 (problem-focused coping strategies) suggests that problem-solving coping strategies may help reduce anxiety, especially on A3 (separation anxiety) and A4 (social phobia).

It is noteworthy that the main effects of specific variables and their interactions with NSSI behaviors show different patterns on various anxiety subscales, revealing the complex relationships between these factors and adolescent anxiety. NSSI and the interaction with parental rearing style: Group-NSSI: the significant positive estimate of F1 on A2, A3, and A4 suggests that perceiving less emotional warmth from fathers is associated with higher anxiety levels in the NSSI group, and this relationship is not as pronounced in the non-NSSI group. This may suggest that NSSI adolescents are more sensitive to the perceived warmth from their fathers, and that its impact on anxiety is more pronounced. Interaction between NSSI and adolescent coping strategies: There were no variables that significantly affected the level of anxiety in the interaction between NSSI and adolescent coping strategies. This

suggests that the impact of adolescent coping strategies on anxiety levels is not influenced by whether or not they have NSSI symptoms.

The analysis of the impact of parental rearing style and personal coping style on anxiety in these two groups of adolescents shows some similarities and differences. The commonality is that positive problem-solving strategies appear to be an effective way to reduce anxiety, regardless of whether the group has NSSI symptoms or not. The difference is that adolescents in the NSSI group are more sensitive to parental rejection, overprotection, and harsh punishment, which are closely related to their higher levels of anxiety. In contrast, in the non-NSSI group, the emotional warmth and understanding of fathers appear to play a more important role in reducing anxiety.

# DISCUSSION

## NSSI and anxious

Many studies have found an association between NSSI and comorbid depression and anxiety (*Hoff & Muehlenkamp, 2009*; *Jacobson et al., 2008*). However, recent research suggests that anxiety may play a greater role than depression in NSSI (*Klonsky, Oltmanns & Turkheimer, 2003*). In this study, we also found through the subscales of the SCARED anxiety scale that adolescents in the NSSI group had higher scores of anxieties in all dimensions (except for separation anxiety (A3)) than those in the non-NSSI group ($p < 0.001$). This highlights the importance of understanding the role of anxiety in NSSI and the need for practitioners to consider anxiety management as a potential treatment focus when working with individuals involved in NSSI.

## NSSI and parenting style

Currently, scholars typically classify parenting styles into the following categories: authoritative (high warmth and high strictness), authoritarian (low warmth and high strictness), permissive (high warmth and low strictness), and neglectful (low warmth and low strictness). According to *Linehan (1993)*, receiving harsh parenting may impact the likelihood of engaging in NSSI behavior. *Victor et al. (2019)* also emphasized that in longitudinal designs, harsh parenting style predicts an increased likelihood of subsequent adolescent NSSI. However, some scholars have pointed out that parenting styles are closely related to the culture of the country and society (*Baetens et al., 2014*; *Baetens et al., 2015*). In China, harsh parenting is usually seen as a sign of parental involvement, care, and love; in Chinese families, fathers often play a more authoritative role and their punishment and severity are usually significant factors influencing NSSI, as shown in Table 1 ($Z = 8.47$, $P = 0.001$).

In the cultural context of Chinese society, traditional family structures and parental roles have a profound impact on children's psychological development. Fathers are usually seen as symbols of authority in the family, and their high-pressure educational methods and authoritarian attitude may lead to strong psychological stress and anxiety in children. In China's collectivist culture, great importance is placed on family harmony and honor, and the authoritative role of the father deeply affects a child's perception of their own identity and worth. However, when this authoritative father role of symbolizing authority

becomes excessively protective or interferes too much with their children, it often leads to adolescent anxiety and stress (*Last, 1993*; *Kerns et al., 2013*), which can also be observed in Table 3 ($P \leq 0.0035$). Positive family interactions and emotional support, especially the understanding and care of the father, are key factors in maintaining children's mental health. Therefore, understanding the impact of the father role on children's mental health is of great importance for preventing and intervening in non-suicidal self-injurious behavior (NSSI).

Compared with the authoritarian parenting style characterized by low warmth and high strictness, we also found that the NSSI group exhibited a lack of parental emotional warmth/understanding and excessive interference behavior ($Z = 13.25$, $P < 0.001$; $Z = 6.74$, $P < 0.001$; $Z = 14.36$, $P < 0.001$; $Z = 6.84$, $P < 0.001$). Therefore, we believe that neglectful and authoritarian parenting styles are also significant influencing factors for NSSI behavior. In China, strict parenting is often viewed as a sign of parental involvement, care, and love, especially within Confucian culture, where high expectations and strict discipline from parents are seen as ways to help children adapt to society (*Chao & Tseng, 2002*; *Lui & Rollock, 2013*). In contrast, in Western cultures, parents are more likely to encourage children's independence, autonomy, and self-esteem (*Ng, Pomerantz & Deng, 2014*). These differing parenting styles reflect the distinct social and cultural expectations of each context, with both approaches having their strengths and limitations depending on the specific family situation and social environment.

### NSSI and adolescent coping strategies

Currently, more and more studies are interested in the impact of coping strategies on NSSI (*Castro & Kirchner, 2018*; *Trepal, Wester & Merchan, 2015*). Coping strategies consist of two parts: active (behavior-oriented) and passive (emotion-oriented). Our study showed that the NSSI group is more inclined towards emotion oriented coping styles ($P < 0.001$), while the non-NSSI group of adolescents tended to have higher scores in behavior-oriented coping strategies ($P < 0.001$). According to *Castro & Kirchner (2018)*, people who display avoidant coping styles are three times more likely to engage in NSSI than those who display approach coping styles. These differences highlight the potential impact of adolescent coping strategy choices on their mental health and behavioral patterns, especially when dealing with stress and emotional challenges. Specifically, *Nock, Prinstein & Sterba (2010)* found that young people who engage in self-harm could delay the engagement in NSSI by using other positive coping strategies, such as distraction or engaging in behavioral coping strategies like talking to someone. Therefore, problem-solving oriented coping strategies may help adolescents manage stress more effectively and reduce the risk of self-harm, while emotion-oriented coping strategies may be associated with higher emotional distress and self-harm risk.

### Parenting style and anxiety

In comparison to the impact of NSSI, we found that permissive parenting style, characterized by high warmth and low demands (overprotection), emerged as a significant predictor of high anxiety levels in this study. As shown in Table 3, in contrast to father's

harsh punishment, we found that mother's harsh punishment emerged as a significant predictor of high anxiety levels. In the cultural context of Chinese society, the role of mothers in the family has a profound impact on the psychological well-being of children. In contrast to the authoritative and punitive parenting style of fathers, the punishment and strict parenting style of mothers was also found to be a significant factor leading to high anxiety levels. A cross-sectional study by *Eun et al. (2018)* similarly showed that different types of parenting styles are closely related to adolescent anxiety symptoms, and the "highly maternal control" parenting style is associated with a higher risk of depression and anxiety.

Traditionally, women were expected to assume the primary responsibility for childcare and education. Influenced by Confucian culture, Chinese women have been assigned the role of "virtuous wives and good mothers" to maintain family harmony and stability. To meet these expectations, mothers often adopt strict and controlling parenting styles to ensure that their children's behavior conforms to social norms and family expectations. In another study, *Aunola & Nurmi (2005)* (*Trepal, Wester & Merchan, 2015*) attempted to emphasize the combination of parental parenting styles, and the results showed that high levels of psychological control implemented by mothers can lead to an increase in children's behavioral problems. High levels of maternal control may limit children's emotional expression and social interaction, and increase levels of emotional distress and anxiety. When mothers adopt over-controlling and punitive strategies, children may feel suppressed and constrained, unable to effectively cope with external pressures and challenges, thereby increasing the risk of anxiety and depression.

Therefore, understanding maternal parenting styles and their impact on children's mental health is of great significance for developing effective psychological interventions. Encouraging mothers to adopt supportive and flexible parenting styles can help reduce adolescent anxiety and depression symptoms and promote mental health and overall development.

## NSSI, parenting style, adolescent coping strategies and anxiety

In the above discussion, we conducted univariate analysis on NSSI, parental parenting styles, adolescent coping strategies, and anxiety emotions separately. However, univariate analysis only focuses on the influence of each independent variable on the results, ignoring the complex relationships that may exist between variables. Interaction analysis can reveal how two or more variables jointly affect the outcome, provide a more comprehensive and accurate understanding, help identify potential moderating effects, and refine subgroup analyses, thereby more effectively guiding actual interventions and measures formulation. Therefore, after screening out significant independent variables through MANOVA, we further adopted regression analysis of main effects and their interaction terms with NSSI to explore in-depth the relationship between these independent variables and specific anxiety factors. These regression models take into account the main effects of parental parenting styles, adolescent coping strategies, and anxiety levels, as well as their interactions with NSSI status.

In Table 3, we further found that NSSI group adolescents' perceived emotional warmth from fathers was significantly associated with their anxiety levels ($P \leq 0.03$), while this

phenomenon was not evident in the non-NSSI group. Specifically, NSSI adolescents have a higher need and sensitivity for emotional warmth from parents. The lack of emotional support and warmth may make it more difficult for them to regulate their emotions, thereby exacerbating their anxiety symptoms. The lack of emotional support and understanding may lead to a decrease in emotional regulation ability among these adolescents, thereby increasing their anxiety symptoms. In traditional Chinese culture, the family is seen as the core of personal emotional support and social function, with fathers typically regarded as the authority and emotional pillar of the family. Therefore, the absence of emotional warmth and understanding from fathers may have a greater negative impact on the mental health of NSSI adolescents.

In contrast, non-NSSI adolescents may be more mature and independent in their emotional regulation and coping mechanisms, and may have relatively less need for emotional warmth and understanding from fathers, or be able to obtain emotional support through other means. This may be because non-NSSI adolescents have higher self-efficacy and independence in coping with stress and emotional distress, and are able to effectively respond to emotional challenges from the outside world, thereby making the impact of father's emotional attitude on their anxiety levels not significant. This conclusion has also been verified to some extent in studies of the tendency of coping styles among adolescents in NSSI and non-NSSI groups (*Trepal, Wester & Merchan, 2015*; *Nock, Prinstein & Sterba, 2010*; *Eun et al., 2018*; *Aunola & Nurmi, 2005*). Meanwhile, *Baetens et al. (2014)* and *Baetens et al. (2015)* found that the NSSI group reported higher levels of parental response control and psychological control. Under high authority and psychological pressure, adolescents with a tendency for NSSI are often in a highly passive position in their family relationships, and their every move is more susceptible to the influence of father's emotional factors, especially in China's patriarchal society.

In Chinese culture, family harmony and parents' high expectations for their children are highly valued. Therefore, by strengthening emotional communication and support from fathers within the family, the anxiety level of adolescents in the NSSI group can be effectively reduced. This means that when developing intervention measures, particular attention should be paid to the role of fathers in emotional interaction, encouraging them to provide more emotional support and understanding to help adolescents better regulate their emotions, relieve anxiety symptoms, and promote their overall psychological well-being. Understanding and improving the emotional interaction between fathers and adolescents is of great significance in relieving anxiety symptoms in NSSI adolescents and promoting their psychological health.

## Strengths & limitations

The strength of this study lies in the comprehensiveness and diversity of the data collected. Multiple validated questionnaires were used, such as the Parenting Style Questionnaire, Coping Strategy Scale, and SCARED Anxiety Scale, to assess the relationship between parenting styles, adolescent coping strategies, and anxiety levels from multiple perspectives. The study used a healthy adolescent control group matched for gender and age frequencies,

which improved the effectiveness of the comparison between the NSSI group and the non-NSSI group and reduced the confounding effect caused by differences in demographic variables. The study had a large sample size, including 311 NSSI patients and 311 healthy controls, which provided sufficient statistical power and enhanced the reliability and generalizability of the study results. In addition, multivariate analysis of variance (MANOVA) and regression analysis were used to evaluate the overall impact of independent variables on dependent variables from multiple dimensions, revealing the interaction between complex variables and providing a more comprehensive understanding.

However, this study also has some limitations. The cross-sectional study design restricted the inference of causal relationships, although associations could be identified, the temporal sequence of events and causal relationships cannot be determined. Relying on self-report questionnaires may introduce response bias, such as social desirability bias or recall bias, which can lead participants to underestimate or overestimate certain behaviors or emotions, and consequently affect the accuracy of the results. The study was conducted only in Ningbo, Zhejiang Province, China, and the results may not be applicable to adolescents from other regions or cultural backgrounds. Future studies should consider including participants from different regions and cultural backgrounds. Without longitudinal data, it is also difficult to determine the temporal changes in the relationships between observed parenting styles, coping strategies, and anxiety levels. Longitudinal studies will provide more information about the stability and evolution of these relationships over time. Despite efforts to control for certain variables, there may still be unmeasured confounding factors, such as socioeconomic status, academic pressure, and peer relationships, which may influence the results. The focus of the study is on hospitalized NSSI adolescents, which may represent a more severe or specific group of NSSI behavior, limiting the generalizability of the results to non-clinical NSSI populations or individuals who do not seek clinical interventions.

In conclusion, this study provides important insights into the relationship between parenting styles, adolescent coping strategies, anxiety levels, and NSSI behaviors through multidimensional data collection and analysis, but caution should be taken in inferring the generalizability and causality of the results. Future studies should consider longitudinal designs and more diverse samples to further validate and extend the findings of this study.

## CONCLUSION

This study reveals significant differences in various variables between NSSI and non-NSSI adolescents by comparing their anxiety levels, parenting styles, and coping strategies. The study found that NSSI adolescents scored significantly higher than non-NSSI adolescents in multiple dimensions of anxiety, such as generalized anxiety, somatic/panic, social phobia, and school phobia. In addition, NSSI adolescents perceive less emotional warmth and understanding from their parents, and experience more punishment, strictness, and interference from them. Moreover, NSSI adolescents tend to use emotion-focused coping strategies rather than problem-focused ones, which is associated with their higher level of anxiety. The results of MANOVA and regression analysis further emphasize the significant

impact of parenting styles and adolescent coping strategies on anxiety levels. Especially, the emotional warmth and understanding from fathers and the punishment and strictness from mothers have significant impacts on anxiety levels in NSSI adolescents. Problem-focused coping strategies can effectively reduce anxiety levels in adolescents, while emotion-focused coping strategies may increase anxiety risk.

This study emphasizes the importance of focusing on emotional support, parenting style, and encouraging positive coping strategies in the family environment in the prevention and intervention of NSSI behavior, through a comprehensive analysis of the effects of NSSI behavior, parenting styles, and adolescent coping strategies on anxiety levels. These findings provide important theoretical basis for the development of effective psychological health interventions. Future research should include longitudinal designs and more diverse cultural backgrounds to further validate and expand these findings, providing more comprehensive guidance for the protection and improvement of adolescent mental health.

### Funding
This work was financially supported by the following grants: Ningbo Public Welfare Research Program (Grant No. 2023S110), the Natural Science Foundation of Ningbo (Grant No. 202003N4262), the Medical and Health Technology of Zhejiang Province (Grant No. 2021KY330), and the Zhejiang Province Traditional Chinese Medicine Science and Technology Plan Project (Grant No. 2020ZB229). The funders had no role in study design, data collection and analysis, decision to publish, or preparation of the manuscript.

### Grant Disclosures
The following grant information was disclosed by the authors:
Ningbo Public Welfare Research Program: 2023S110.
Natural Science Foundation of Ningbo: 202003N4262.
Medical and Health Technology of Zhejiang Province: 2021KY330.
Zhejiang Province Traditional Chinese Medicine Science and Technology Plan Project: 2020ZB229.

### Competing Interests
The authors declare there are no competing interests.

### Author Contributions
- Lingjiang Liu performed the experiments, analyzed the data, prepared figures and/or tables, authored or reviewed drafts of the article, and approved the final draft.
- Xinhui Hu conceived and designed the experiments, authored or reviewed drafts of the article, and approved the final draft.
- Huabing Xie conceived and designed the experiments, authored or reviewed drafts of the article, and approved the final draft.

- Changzhou Hu performed the experiments, prepared figures and/or tables, and approved the final draft.
- Dongsheng Zhou analyzed the data, authored or reviewed drafts of the article, and approved the final draft.
- Jie Zhang performed the experiments, prepared figures and/or tables, and approved the final draft.
- Yangjian Kong performed the experiments, prepared figures and/or tables, authored or reviewed drafts of the article, and approved the final draft.
- Fang Cheng conceived and designed the experiments, analyzed the data, prepared figures and/or tables, and approved the final draft.

**Human Ethics**

The following information was supplied relating to ethical approvals (*i.e.*, approving body and any reference numbers):

This survey was conducted following the principles of the Declaration of Helsinki. The following information was supplied relating to ethical approvals (*i.e.*, approving body and any reference numbers):

The Ningbo University Affiliated Kangning Hospital granted Ethical approval to carry out the study within its facilities (Ethical Application Ref. No.: NBKNYY-2019-LC-10, 2019.7.25-2022.7.24).

**Data Availability**

The raw measurements and analysis code are available in the Supplementary Files.

**Supplemental Information**

Supplemental information for this article can be found online at http://dx.doi.org/10.7717/peerj.18378#supplemental-information.

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
