# Peer review of "The influence of parenting styles and coping strategies on anxiety symptoms in adolescents: a comparative study of groups with and without non-suicidal self-injury behavior"

_PeerJ, doi:10.7717/peerj.18378_

## Round 0.1 · original submission · Major Revisions

Dear Dr. Cheng,

Thank you for your submission to PeerJ. After review by two reviewers, your article requires some revision: Major Revisions. Both reviewers agree that Major Revisions are necessary, please review the changes suggested by the reviewers point by point and note that Reviewer 1 has provided an annotated manuscript as a PDF attachment.

Thank you.

·

Basic reporting

There is no consistency in the focus of the discussion between the abstract discussing globally, the introduction discussing Covid 19, and the discussion discussing Chinese culture.
if the cultural background is to be underlined. In the method, especially the population and sample, the characteristics of the respondents are written, such as native Chinese parents with a Chinese mindset.

Experimental design

no comment

Validity of the findings

No Comment

Additional comments

Parenting style is influenced by culture; this needs to be connected at the beginning of the introduction and abstract. so that it becomes synchronized from the abstract, introduction, and discussion. did not suddenly appear in the discussion section.
culture is more suitable for use in background and introduction.

Reviewer 2 ·

Basic reporting

Thank you for inviting me to review this paper, which explores the influence of parenting styles and coping strategies on anxiety symptoms in adolescents, specifically comparing groups with and without nonsuicidal self-injury behavior. The manuscript is concise and understandable. Here are some suggestions for improvement:

1、Please rewrite the abstract to make it more engaging and less lengthy.
2、Consider shortening and refining the Introduction and Discussion sections to enhance clarity and focus.
3、Try to incorporate more recent high-level articles into the references to strengthen the paper's academic standing.

Experimental design

Research question well defined, relevant & meaningful.

Validity of the findings

Conclusions are well stated,

---

## Round 0.2 · accepted · Accept

Dear Dr. Cheng:

Thank you for submitting your article to PeerJ. After review by the peer reviewers, your article has been accepted.

Sincerely
Ana María Jiménez-Cebrián

Reviewer 2 ·

Basic reporting

yes

Experimental design

yes

Validity of the findings

yes

Additional comments

none